# EVALUATING DEEP GRAPH NEURAL NETWORKS

## ABSTRACT

Graph Neural Networks (GNNs) have already been widely applied in various graph mining tasks. However, most GNNs only have shallow architectures, which limits performance improvement. In this paper, we conduct a systematic experimental evaluation on the fundamental limitations of current architecture designs. Based on the experimental results, we answer the following two essential questions: (1) what actually leads to the compromised performance of deep GNNs; (2) how to build deep GNNs. The answers to the above questions provide empirical insights and guidelines for researchers to design deep GNNs. Further, we present Deep Graph Multi-Layer Perceptron (DGMLP), a powerful approach implementing our proposed guidelines. Experimental results demonstrate three advantages of DGMLP: 1) high accuracy – it achieves state-of-the-art node classification performance on various datasets; 2) high flexibility – it can flexibly choose different propagation and transformation depths according to certain graph properties; 3) high scalability and efficiency – it supports fast training on large-scale graphs.

## 1 INTRODUCTION

The recent success of Graph Neural Networks (GNNs) (Zhang et al., 2020) has boosted researches on knowledge discovery and data mining on graph data. Designed for graph-structured data, GNNs provide a universal way to tackle node-level, edge-level, and graph-level tasks, including social network analysis (Qiu et al., 2018; Fan et al., 2019; Huang et al., 2021), chemistry and biology (Dai et al., 2019; Bradshaw et al., 2019; Do et al., 2019), recommendation (Monti et al., 2017; Wu et al., 2020; Yin et al., 2019), natural language processing (Bastings et al., 2017; Wu et al., 2021; Vashishth et al., 2020), and computer vision (Qi et al., 2018; Shi et al., 2019; Sarlin et al., 2020).

The key to the success of most GNNs lies in the graph convolution operation, which propagates neighbor information to the center node in an iterative manner (Wu et al., 2019). The graph convolution operation can be further decomposed into two sequential operations: *embedding propagation* (EP) and *embedding transformation* (ET). The EP operation can be viewed as a special form of Laplacian smoothing (NT & Maehara, 2019), which combines the embeddings of a node itself and its one-hop neighbors. The embeddings of nodes within the same connected component would become similar after applying the smoothing operation, which greatly eases the downstream tasks. The ET operation applies neural networks and transforms the node embeddings to target dimensions. Taking the widely-used Graph Convolutional Network (GCN) (Kipf & Welling, 2016) as an example, through stacking $k$ convolution operations (i.e., layers), each node in GCN can utilize the information from nodes within its $k$-hop neighborhood, and thus improve the predictive accuracy by getting more unlabeled nodes involved in the training process.

Despite the remarkable success, simply stacking many graph convolution operations leads to massive performance degradation. As a result, most GNNs today only have shallow architectures (e.g., 2 or 3 layers), which limits their exploitation of deep structural information. Concretely, under the semi-supervised setting where only a few labels are given, shallow GNNs can utilize only a small percentage of nodes during model training, leading to sub-optimal node classification accuracy.

To alleviate the problem that GNNs cannot go deep, many researches have been proposed, and they attribute the performance degradation of deep GNNs to several reasons. Among the suggested reasons, most existing works (Feng et al., 2020; Chen et al., 2020a; Zhao & Akoglu, 2020; Godwin et al., 2021; Rong et al., 2019; Zeng et al., 2020a; Min et al., 2020; Chamberlain et al., 2021; Chien et al., 2021; Zhou et al., 2020a; Hou et al., 2019; Beaini et al., 2021; Yan et al., 2021; Cai & Wang,

2020) consider the *over-smoothing* issue as the major cause of performance degradation of deep GNNs. Notice that the EP operation smooths the node embeddings, i.e., making nodes within the same connected component similar. If a GNN is stacked with a large number of graph convolution operations, the output embeddings might be over-smoothed, i.e., nodes within the same connected component become indistinguishable.

**Questions Investigated.** In this paper, we dive deep into the problem of why most existing GNNs cannot go deep and try to present answers to the following two key questions:

**Q1**: What actually limits the deep stacking of convolution operations in GNN designs?

**Q2**: How can we design deep GNNs with the help of the findings from the experimental analysis and outperform the state-of-the-art GNNs?

**Contributions.** To answer the above research questions, we first conduct a comprehensive evaluation to revise the *over-smoothing* issue and identify the root cause of performance degradation of most existing GNNs when they go deep. Based on the above analysis, we obtain helpful insights and guidelines to design deep GNNs. Our main contributions are summarized as follows.

**C1**: We clarify the concept of model depth by separating and considering the two different depths when designing deep GNNs: *the propagation depth $D_p$* and *the transformation depth $D_t$*. Through experimental evaluations, we find that large $D_p$ leads to the *over-smoothing* issue whereas large $D_t$ leads to the *model degradation* issue in the current GNN models. Moreover, we observed that the latter usually happens much earlier than the former as $D_p$ and $D_t$ increase at the same speed. Thus, the *model degradation* issue introduced by large $D_t$ is the true root cause for the failure of deep GNNs.

**C2**: To design models that support large $D_p$, we propose a node-adaptive combination mechanism for combining propagated features under EP operations of different steps. To support large $D_t$, we add residual connections between ET operations to alleviate the *model degradation* issue. Further, we present Deep Graph Multi-Layer Perceptron (DGMLP), a novel approach that adopts the composition of the above mentioned two mechanisms to successfully support both large $D_p$ and large $D_t$ based on our findings from the experimental analysis. We validate the effectiveness of DGMLP on six public datasets and the Industry dataset from the real industrial environment. Experimental results demonstrate that DGMLP outperforms the SOTA GNNs while maintaining high scalability and efficiency.

To the best of our knowledge, this paper is the first to conduct an experimental evaluation that identifies the major reason why most existing GNNs cannot go deep. Our findings and the derived guidelines open up a new perspective on designing deep GNNs for graph-structured data.

## 2 PRELIMINARY

In this section, we first explain the problem formulation. Then we introduce *Embedding Propagation* (EP) and *Embedding Transformation* (ET) in the graph convolution operation in detail.

### 2.1 PROBLEM FORMALIZATION

In this paper, we consider an undirected graph $\mathcal{G} = (\mathcal{V}, \mathcal{E})$ with $|\mathcal{V}| = N$ nodes and $|\mathcal{E}| = M$ edges. $\mathbf{A}$ is the adjacency matrix of $\mathcal{G}$, weighted or not. Each node possibly has a feature vector of size $d$, stacked up to an $N \times d$ matrix $\mathbf{X}$. $\mathbf{D} = \mathrm{diag}(d_1, d_2, \cdots, d_n) \in \mathbb{R}^{N \times N}$ denotes the degree matrix of $\mathbf{A}$, where $d_i = \sum_{j \in \mathcal{V}} \mathbf{A}_{ij}$ is the degree of node $i$. In this paper, we focus on the semi-supervised node classification task. Suppose $\mathcal{V}_l$ is the labeled node set, the goal is to predict the labels for nodes in the unlabeled set $\mathcal{V}_u$ under the limited supervision of labels for nodes in $\mathcal{V}_l$.

### 2.2 CONVOLUTION ON GRAPHS

**Graph Convolution.** Based on the intuitive assumption that locally connected nodes are likely to have the same label (McPherson et al., 2001), GNN iteratively propagates the information of each

Figure 1: The relationship between GCN and MLP.

node to its adjacent nodes. For example, each graph convolution operation in GCN firstly propagates the node embeddings to their neighborhoods and then transforms their propagated node embeddings:

$$\mathbf{X}^{(k+1)} = \sigma\big(\hat{\mathbf{A}}\mathbf{X}^{(k)}\mathbf{W}^{(k)}\big), \qquad \hat{\mathbf{A}} = \widetilde{\mathbf{D}}^{\frac{1}{2}}\widetilde{\mathbf{A}}\widetilde{\mathbf{D}}^{-\frac{1}{2}}, \tag{1}$$

where $\mathbf{X}^{(k)}$ and $\mathbf{X}^{(k+1)}$ are the node embedding matrices at layer $k$ and $k+1$, respectively. $\widetilde{\mathbf{A}} = \mathbf{A} + \mathbf{I}_N$ is the adjacency matrix of the undirected graph $\mathcal{G}$ with self loops added, where $\mathbf{I}_N$ is the identity matrix. $\hat{\mathbf{A}}$ is the normalized adjacency matrix, and $\widetilde{\mathbf{D}}$ is its corresponding degree matrix.

By setting different $r$ in $\hat{\mathbf{A}} = \widetilde{\mathbf{D}}^{r-1}\widetilde{\mathbf{A}}\widetilde{\mathbf{D}}^{-r}$, different normalization strategies can be employed, such as the symmetric normalized matrix $\widetilde{\mathbf{D}}^{-\frac{1}{2}}\widetilde{\mathbf{A}}\widetilde{\mathbf{D}}^{-\frac{1}{2}}$ (Kipf & Welling, 2016), the random walk transition probability matrix $\widetilde{\mathbf{D}}^{-1}\widetilde{\mathbf{A}}$ (Xu et al., 2018), and the reverse random walk transition probability matrix $\widetilde{\mathbf{A}}\widetilde{\mathbf{D}}^{-1}$ (Zeng et al., 2020b). We adopt $\hat{\mathbf{A}} = \widetilde{\mathbf{D}}^{-\frac{1}{2}}\widetilde{\mathbf{A}}\widetilde{\mathbf{D}}^{-\frac{1}{2}}$ in this work.

**EP and ET Operations.** Each graph convolution operation in GNNs can be decomposed into two sequential operations: *Embedding Propagation* (EP) and *Embedding Transformation* (ET). This decomposition naturally leads to two corresponding GNN depths: propagation depth $D_p$ and transformation depth $D_t$. Concretely, GNN first executes EP, which generates smoothed features by multiplying the normalized adjacency matrix $\hat{\mathbf{A}}$ with the node embedding matrix $\mathbf{X}$:

$$\text{EP}(\mathbf{X}) = \hat{\mathbf{A}}\mathbf{X}. \tag{2}$$

Then, the smoothed features $\hat{\mathbf{X}} = \text{EP}(\mathbf{X})$ will be transformed with the learnable transformation matrix $\mathbf{W}$ and the activation function $\sigma(\cdot)$:

$$\text{ET}(\hat{\mathbf{X}}) = \sigma(\hat{\mathbf{X}}\mathbf{W}). \tag{3}$$

Fig. 1 shows the framework of a two-layer GCN. To note that, GCN will degrade to MLP if $\hat{\mathbf{A}}$ is the identity matrix, which is equal to removing the EP operation in all GCN layers. More detailed analysis and classification of current GNN approaches can be found in Appendix A.

## 3  SMOOTHNESS MEASUREMENT

In Eq. 2, each time $\hat{\mathbf{A}}$ multiplies with $\mathbf{X}$, information one more hop away can be acquired for each node. Thus, in order to fully leverage high-order neighborhood information, a series of multiplications of $\hat{\mathbf{A}}\mathbf{X}$, i.e., the EP operation, have to be carried out, which means stacking multiple GNN layers. However, if we execute $\hat{\mathbf{A}}\mathbf{X}$ numerous times, the node embeddings within the same connected component would reach a stationary state, leading to indistinguishable node embeddings (i.e., over-smoothing issue). Concretely, when adopting $\hat{\mathbf{A}} = \widetilde{\mathbf{D}}^{r-1}\tilde{\mathbf{A}}\widetilde{\mathbf{D}}^{-r}$, $\hat{\mathbf{A}}^{\infty}$ follows

$$\hat{\mathbf{A}}^{\infty}_{i,j} = \frac{(d_i+1)^r(d_j+1)^{1-r}}{2m+n}, \tag{4}$$

which shows that after infinite times of multiplication, the influence from node $i$ to $j$ is only determined by the degrees of them. Under this scenario, the neighborhood information is fully corrupted, resulting in catastrophic node classification accuracy.

As the over-smoothing issue is only introduced by the EP operation rather than the ET operation, here we introduce a new metric, "Node Smoothness Level ($NSL$)", to evaluate the smoothness of a node after $k$ steps of EP operation. Suppose $\mathbf{X}^{(0)} = \mathbf{X}$ is the original node feature matrix, and $\mathbf{X}^{(k)} = \hat{\mathbf{A}}^k\mathbf{X}^{(0)}$ is the smoothed features after $k$ times of EP operation.

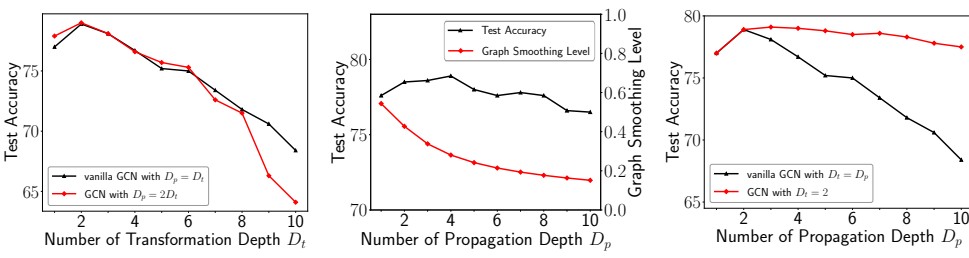

(a) The influence of $D_p$ to model performance.
(b) The influence of $GSL$ to model performance.
(c) The influence of $D_t$ to model performance.

Figure 2: Over-smoothing is not the main contributor who hurts the performance of deep GNNs.

**Definition 3.1** (**Node Smoothing Level**). *The Node Smoothing Level $NSL_v(k)$ parameterized by node $v$ and the EP steps, $k$, is defined as:*

$$\alpha = Sim(\mathbf{x}_v^k, \mathbf{x}_v^0), \quad \beta = Sim(\mathbf{x}_v^k, \mathbf{x}_v^\infty), \quad NSL_v(k) = \alpha * (1 - \beta), \tag{5}$$

*where $\mathbf{x}_v^k$ is the smoothed feature of node $v$ after $k$ steps of EP operation, $\mathbf{x}_v^0$ represents node $v$'s original feature, and $\mathbf{x}_v^\infty$ represents node $v$'s feature at stationary state. $Sim(\cdot)$ is a similarity function, being the cosine similarity in the following discussion.*

Further, the "Graph Smoothing Level" ($GSL$) parameterized by the EP steps, $k$, is defined as:

$$GSL(k) = \frac{1}{N} \sum_{v \in \mathcal{V}} NSL_v(k). \tag{6}$$

Smaller $GSL(k)$ means that $\mathbf{X}^{(k)}$ is more likely to forget the original node feature information $\mathbf{X}^{(0)}$ after $k$ steps of EP operation and has a higher risk of the *over-smoothing* issue.

## 4 MISCONCEPTIONS AND THE TRUE ROOT CAUSE

Most previous works (Li et al., 2018; Zhang et al., 2019) claim that the *over-smoothing* issue is main cause for the failure of deep GNNs. There have been lines of works that aim at designing deep GNNs. For example, DropEdge (Rong et al., 2019) randomly removes edges during training, and Grand (Feng et al., 2020) randomly drops raw features of nodes before propagation. Despite their ability to go deeper while maintaining or even getting better predictive accuracy, the explanations for their effectiveness are misleading in some instances. The experimental analysis about misconceptions other than the *over-smoothing* issue can be found in Appendix C.

### 4.1 IS OVER-SMOOTHING REALLY THE ROOT CAUSE?

**Enlarging $D_p$ in Vanilla GCN.** To investigate the relations between smoothness and node classification accuracy, we increase the number of graph convolutional layers in vanilla GCN ($D_p = D_t$) and a modified GCN with $\hat{\mathbf{A}}^2$ being the normalized adjacency matrix ($D_p = 2D_t$) on the PubMed dataset (Sen et al., 2008). Supposing that the *over-smoothing* issue is the main cause for the failure of deep GNNs, the predictive accuracy of the GCN with $D_p = 2D_t$ should be way lower than the one of vanilla GCN. The experimental results are shown in Fig. 2(a).

From Fig. 2(a), we can see that even with a higher level of smoothness, GCN with $D_p = 2D_t$ always has similar predictive accuracy with vanilla GCN ($D_p = D_t$) when $D_t$ ranges from 1 to 8, and the *over-smoothing* issue seems to begin dominating the performance decline only when $D_p$ exceeds 16 ($2 \times 8$). The performance of vanilla GCN does decrease sharply when $D_p$ exceeds 2, which is precisely the situation the *over-smoothing* issue suggests. However, even with relatively large $D_p$ (e.g., 12), the predictive accuracy of the model with larger smoothness (GCN with $D_p = 2D_t$) is similar to the vanilla GCN, which on the contrary implies that the *over-smoothing* issue may not be the major cause for performance degradation of deep GNNs until the graph smoothness achieves an extremely high level (e.g., $D_p > 16$ on the PubMed dataset).

**Enlarging $D_p$ in SGC.** To further validate our guess, we increase the number of propagation depth $D_p$ of SGC and then evaluate the corresponding predictive accuracy and the value of $GSL$ defined

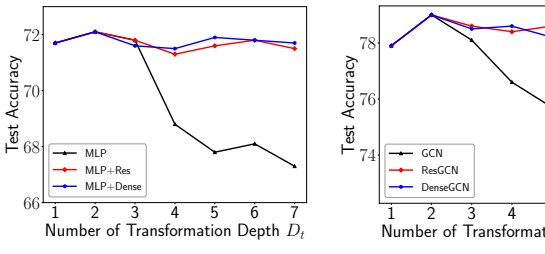

(a) The skip connection to MLP.   (b) The skip connection to GCN.

Figure 3: Performance comparison when adding Residual and Dense connection.

in Sec. 3. We present the evaluation results in Fig. 2(b). By increasing $D_p$ from 1 to 10, the value of $GSL$ has decreased by more than 60%, but the corresponding predictive accuracy decline is less than 1%. This sharp contrast strongly illustrates that low $GSL$, i.e., the *over-smoothing* issue, is not the main cause for the performance degradation of deep GNNs. Moreover, compared with 10-layer vanilla GCN in Fig. 2(a), the corresponding predictive accuracy of 10-layer SGC is still quite high even with the same $D_p$ as vanilla GCN. Therefore, we further guess that the large $D_t$ may be the root cause for the performance degradation of deep vanilla GCNs.

**Large $D_t$ Dominates Performance Degradation.**   To dig out the true limitation of deep GCNs, we fix the number of transformation depth $D_t$ to 2 and set the normalized adjacency matrix to $\hat{\mathbf{A}}^{D_p/2}$ (when $D_p$ is odd, use $\hat{\mathbf{A}}^{\lfloor D_p/2 \rfloor +1}$ in the first layer and $\hat{\mathbf{A}}^{\lfloor D_p/2 \rfloor}$ in the second layer), and then report the accuracy along with the increased propagation depth $D_p$. The experimental results in Fig. 2(c) shows that the accuracy of GCN with $D_t = 2$ does not drop quickly when $D_p$ becomes large, while it faces a sharp decline in vanilla GCN, which fixes $D_p = D_t$. Individually enlarging $D_p$ will increase the risk of the *over-smoothing* issue, but the accuracy is only slightly influenced. However, the performance of vanilla GCN experiences a drastic drop if we simultaneously increase $D_t$.

**Findings 1:  Large $D_p$ will harm the predictive accuracy of deep GNN, yet the accuracy decline is relatively small. On the contrary, large $D_t$ is the root cause for the failure of deep GNNs.**

## 4.2   WHAT'S BEHIND LARGE $D_t$?

To learn what is the fundamental problem caused by large $D_t$, we first evaluate the predictive accuracy of deep MLP on the PubMed dataset and then move the research object to deep GNN.

**Deep MLP Also Performs Bad.**   We evaluate the predictive accuracy of MLP along with $D_t$, i.e., the number of MLP layers, on the PubMed dataset, and the black line in Fig. 3(a) shows the evaluation results. It can be easily drawn from the results that the predictive accuracy of MLP also decreases sharply when $D_t$ increases. Thus, the performance degradation caused by large $D_t$ also exists in MLP. It reminds us that the approaches easing the training of deep MLP might also help alleviate the performance degradation caused by large $D_t$ in GNN.

**Skip Connections Can Help.**   The widely-used approach that eases the training of deep MLP is to add skip connections between layers (He et al., 2016; Huang et al., 2017). Here, we add residual and dense connections to MLP and generate two MLP variants: "MLP+Res" and "MLP+Dense", respectively. The accuracy of these two models with increasing $D_t$ is shown in Fig. 3(a). Compared with plain deep MLP, the accuracy of both "MLP+Res" and "MLP+Dense" does not encounter huge degradation when $D_t$ increases. The results illustrate that adding residual or dense connections can effectively alleviate the performance degradation issue cause by large $D_t$.

**Model Degradation.**   The skip connections are first introduced in (He et al., 2016) to alleviate the model degradation issue, which is a phenomenon that the accuracy firstly increases and then decreases rapidly when increasing the number of layers in one model. Surprisingly, the degradation is not caused by overfitting as the training error becomes higher when adding more layers in the model. Adopting the same approach to alleviate the model degradation issue, we add residual and dense connections to GCN and generate two GCN variants: "ResGCN" and "DenseGCN", respectively. The accuracy results in Fig. 3(b) illustrate that the performance decline of both "ResGCN" and "DenseGCN" can be ignored compared to the huge accuracy decline of vanilla GCN.

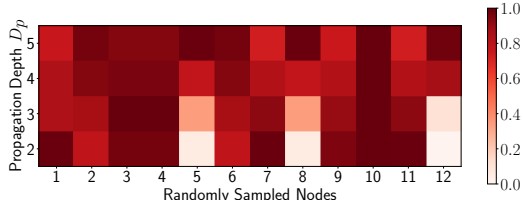

Figure 4: Different nodes reach their optimal performance at varied propagation depth $D_p$.

**Findings 2: The *model degradation* issue behind large $D_t$ is the true root cause for the failure of deep GNNs. And adding skip connections between layers can effectively alleviate the performance degradation issue of deep GNNs.**

## 5 GUIDELINES ON CONSTRUCTING DEEP GNNS

In this section, we propose several guidelines on how to construct deep GNNs that support large propagation depth $D_p$ and large transformation depth $D_t$ based on the experimental analysis in Sec. 4. Further Discussions about when to adopt deep GNNs can be found in Appendix B.

### 5.1 A MORE FLEXIBLE FRAMEWORK

Many recent GNN works follow the framework design proposed by SGC (Wu et al., 2019) which decouples the EP and ET operations inside each GNN layer. The decoupled design split the framework into two components. In the most popular decoupled GNN design, the first component executes the EP operation in a certain manner to generated propagated node features. Then the propagated features are then fed into the second component to execute ET operations. The second component is usually a plain MLP. There are several other methods (Klicpera et al., 2018; Liu et al., 2020) that exchange the order of the two components mentioned above. Under the decoupled framework, the choices of $D_p$ and $D_t$ are more flexible as it breaks the limit that $D_p = D_t$. Thus, for different kinds of graph-structured data, the decoupled framework is able to adopt different values of $D_p$ and $D_t$ for optimal predictive accuracy. For example, $S^2$GC (Zhu & Koniusz, 2021) and GBP (Chen et al., 2020b) first execute the EP operations to generate propagated features at different $D_p$. Then they adopt a heuristic weighting mechanism to combine these propagated features. Finally, the combined features are fed into a plain MLP to get the prediction results.

**Guidelines 1: The decoupled framework should be adopted to free the choices of $D_p$ and $D_t$ from the restraint that $D_p = D_t$ in order to adapt to the characteristics of different datasets.**

### 5.2 HOW TO CONSTRUCT GNNS WITH LARGE $D_p$?

Despite the effectiveness of previous works, a problem still exists: when combining propagated features, the weighting mechanism works at the graph level rather than at the node level. For example, the weighting mechanism in $S^2$GC (Zhu & Koniusz, 2021) and GBP (Chen et al., 2020b) is sub-optimal as it assigns the same weight distribution to all the nodes when combining propagated features at different propagation steps. As a result, the individual properties of each node are ignored. To verify our claim, we apply SGC for the node classification task with different propagation depths on 12 randomly selected nodes in the Cora dataset. We run SGC 100 times and report the average accuracy of the selected nodes. We observe from Fig. 4 that the optimal propagation depths for the selected nodes are highly diverse. The results demonstrate that different nodes should have different weight distributions along with $D_p$ to get the optimal predictive accuracy.

**Guidelines 2: A node-adaptive weighting mechanism should be adopted to satisfy each node's diverse needs for the propagation depth $D_p$ when constructing deep GNNs.**

### 5.3 HOW TO CONSTRUCT GNNS WITH LARGE $D_t$?

Recently, lines of works have been proposed to support large $D_t$, and many of them add skip connections between GNN layers motivated by ResNet (He et al., 2016) and DenseNet (Huang et al., 2017). For example, JK-Net (Xu et al., 2018) proposes a new transformation scheme for node embeddings that combines all node embeddings at previous layers in the final layer. GCNII (Chen

et al., 2020c) addresses small $D_t$ via initial residual connections and identity mappings. Besides, as shown in Fig. 3(b), vanilla GCN with residual or dense connections is also able to support large $D_t$.

**Guidelines 3: Adding skip connections between GNN layers is an effective way for GNN models to support large $D_t$.**

## 6 ONE ALTERNATIVE SOLUTION

Under the guidance of the above guidelines, we propose a scalable and flexible model termed Deep Graph Multi-Layer Perceptron (DGMLP), which contains a node-adaptive weighting mechanism for large $D_p$ and the residual connections for large $D_t$. Following the decoupled framework, DGMLP first calculates the propagated features at different $D_p$ using the EP operation. Then a novel node-adaptive weighting mechanism is proposed to combine the propagated features at different $D_p$ effectively. Finally, the combined feature is fed into an MLP with added skip connections to support large $D_t$. The remainder of this section will introduce the node-adaptive weighting mechanism and the skip connections adopted in DGMLP in detail.

### 6.1 NODE-ADAPTIVE WEIGHTING MECHANISM

After generating the propagated features $\mathbf{X}^{(k)} = \hat{\mathbf{A}}^k \mathbf{X}$ for propagation depth $k$ ranges from 1 to $K$ using the EP operation, we further calculate the $NSL_v(k)$ parameterized by node $v$ and propagation depth $k$ defined in Def. 3.1. Remember that smaller $NSL_v(k)$ means that the node embedding at propagation step $k$ is more likely to forget the original node feature information and has a higher risk of the *over-smoothing* issue. Thus, in this case, propagated feature at propagation step $k$, $\mathbf{x}_v^k$ should be intuitively assigned with smaller weights. To restrain the weights in between 0 and 1, the *propagation weight* $w_v(k)$ parameterized by node $v$ and propagation step $k$ is defined as the softmax output of $\{NSL_v(0), NSL_v(1), \cdots, NSL_v(K)\}$:

$$w_v(k) = \frac{e^{NSL_v(k)/T}}{\sum\limits_{l=0}^{K} e^{NSL_v(l)/T}}. \tag{7}$$

Similar to Knowledge Distillation (Hinton et al., 2015; Lan et al., 2018), the temperature $T$ is adopted here to soften or harden the probability distributions. Smaller $T$ will harden the distributions, and thus the model will focus more on the local graph information.

Finally, the propagated features at different $D_p$ are combined using the weight $w_v(k)$ in Eq. 7 to generate the combined feature $\hat{\mathbf{x}}_v = \sum\limits_{k=0}^{K} w_v(k)\mathbf{x}_v^k$. By adaptively assigning different propagation weights for different nodes, we can simply increase $D_p$ on the graph level and get more powerful node embeddings with personalized smoothing levels for each node.

### 6.2 SKIP CONNECTIONS

Following guidelines 3, we choose to add residual connections (He et al., 2016) between the layers in the MLP of our DGMLP. We refer to the layers in the MLP of our DGMLP as the following format:

$$\mathbf{h}_v^{(l+1)} = \sigma(\mathbf{h}_v^{(l)}\mathbf{W}^{(l)}) + \mathbf{h}_v^{(l)}, \tag{8}$$

where $\mathbf{W}^{(l)}$ is the learnable parameter matrix, $\mathbf{h}_v^{(0)} = \hat{\mathbf{x}}_v$ is the original combined node feature vector, and $\mathbf{h}_v^{(l)}$ is the transformed node embeddings at the $l$-th layer of the MLP with residual connections.

## 7 DGMLP EVALUATION

In this section, we conduct extensive experiments to evaluate our proposed DGMLP. We first introduce the utilized datasets and experiment setup. Then, we compare DGMLP with state-of-the-art baselines in predictive accuracy, scalability, and model depth. More experimental results about efficiency, graph sparsity, and interpretability of DGMLP can be found in Appendix E.

Table 1: Test accuracy on the node classification task. "OOM" means "out of memory".

| Methods | Cora | Citeseer | PubMed | Industry | ogbn-arxiv | ogbn-products | ogbn-papers100M |
|---|---|---|---|---|---|---|---|
| GCN | 81.8±0.5 | 70.8±0.5 | 79.3±0.7 | 45.9±0.4 | 71.7±0.3 | OOM | OOM |
| GraphSAGE | 79.2±0.6 | 71.6±0.5 | 77.4±0.5 | 45.7±0.6 | 71.5±0.3 | 78.3±0.2 | 64.8±0.4 |
| JK-Net | 81.8±0.5 | 70.7±0.7 | 78.8±0.7 | 47.2±0.3 | 72.2±0.2 | OOM | OOM |
| ResGCN | 81.2±0.5 | 70.8±0.4 | 78.6±0.6 | 45.8±0.5 | 72.6±0.4 | OOM | OOM |
| APPNP | 83.3±0.5 | 71.8±0.5 | 80.1±0.2 | 46.7±0.6 | 72.0±0.3 | OOM | OOM |
| AP-GCN | 83.4±0.3 | 71.3±0.5 | 79.7±0.3 | 46.9±0.7 | 71.9±0.2 | OOM | OOM |
| DAGNN | 84.4±0.5 | 73.3±0.6 | 80.5±0.5 | 47.1±0.6 | 72.1±0.3 | OOM | OOM |
| SGC | 81.0±0.2 | 71.3±0.5 | 78.9±0.5 | 45.2±0.3 | 71.2±0.3 | 75.9±0.2 | 63.2±0.2 |
| SIGN | 82.1±0.3 | 72.4±0.8 | 79.5±0.5 | 46.3±0.5 | 71.9±0.1 | 76.8±0.2 | 64.2±0.2 |
| S²GC | 82.7±0.3 | 73.0±0.2 | 79.9±0.3 | 46.6±0.6 | 71.8±0.3 | 77.1±0.1 | 64.7±0.3 |
| GBP | 83.9±0.7 | 72.9±0.5 | 80.6±0.4 | 46.9±0.7 | 72.2±0.2 | 77.7±0.2 | 65.2±0.3 |
| DGMLP | **84.6±0.6** | **73.4±0.5** | **81.2±0.6** | **47.6±0.7** | **72.8±0.2** | **78.5±0.2** | **65.7±0.2** |

## 7.1 EXPERIMENTAL SETTINGS

**Datasets.** We adopt the three popular citation network datasets (Cora, Citeseer, PubMed) (Sen et al., 2008), three large OGB datasets (ogbn-arxiv, ogbn-products, ogbn-papers100M) (Hu et al., 2020), and one Industry dataset from our industrial cooperative enterprise in our evaluation. Table 7 in Appendix D.1 presents an overview of these seven datasets.

**Baselines.** We choose the following baselines: GCN (Kipf & Welling, 2016), GraphSAGE (Hamilton et al., 2017), JK-Net (Xu et al., 2018), ResGCN (Li et al., 2019), APPNP (Klicpera et al., 2018), AP-GCN (Spinelli et al., 2020), DAGNN (Liu et al., 2020), SGC (Wu et al., 2019), SIGN (Frasca et al., 2020), S²GC (Zhu & Koniusz, 2021), and GBP (Chen et al., 2020b). The hyperparameter details for our DGMLP and all the baseline methods can be found in Appendix D.2.

## 7.2 END-TO-END COMPARISON

The classification results on three citation networks are shown in Table 1. We observe that DGMLP outperforms all the compared baseline methods. Notably, the predictive accuracy of DGMLP exceeds the one of current state-of-the-art method GBP by a margin of 0.6% on the largest citation networks dataset, PubMed. Compared with coupled methods (e.g., GCN, JK-Net), the decoupled methods (e.g., DAGNN, GBP) get better predictive accuracy. It is due to the fact that the disentanglement of EP and ET operations enables $D_p$ to go extremely deep, exploiting more deep structural information.

We further evaluate DGMLP on the three large OGB datasets and one Industry dataset, and the results are also summarized in Table 1.

As shown in Table 1, DGMLP consistently achieves the best performance across the four large datasets. The improvement of DGMLP over baseline methods mainly relies on its support of both large $D_p$ and large $D_t$.

## 7.3 TRAINING SCALABILITY

To test the scalability of DGMLP, we use the Erdős-Rényi graph (Erdos et al., 1960) generator in the Python package NetworkX (Hagberg et al., 2008) to generate artificial graphs of different sizes. The node sizes of the generated artificial graphs vary from 0.1 million to 1 million, and the probability of an edge exists between two nodes is set to 0.0001. We choose two representative methods GCN and APPNP as compared baselines. The total running time (including the pre-processing time) of training for 200 epochs and the GPU memory requirement are shown in Fig. 5(a) and Fig. 5(b), respectively. The running time speedup of DGMLP against GCN is also included in Fig. 5(a).

The experimental results in Fig. 5(a) illustrate that DGMLP is highly efficient compared to GCN and APPNP. It only takes DGMLP 223.4 seconds to finish the training on a large graph of size 1 million, which is less than the running time of both GCN and APPNP on the graph of size 0.3 million. Fig. 5(b) shows that the GPU memory requirement of DGMLP grows almost linearly along with graph size. On the contrary, the GPU memory requirements of GCN and APPNP both grow much quicker than DGMLP, exceeding 16GB when the graph size is 1 million, while the memory requirement of DGMLP is just over 3GB at the same graph size. It indicates that our proposed DGMLP enjoys high scalability and high efficiency at the same time.

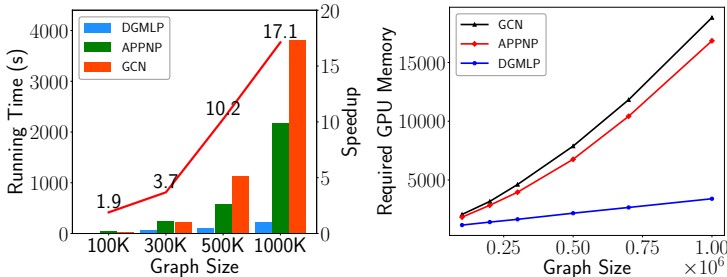

Figure 5: Running time and GPU memory requirement comparison on different sizes of graphs

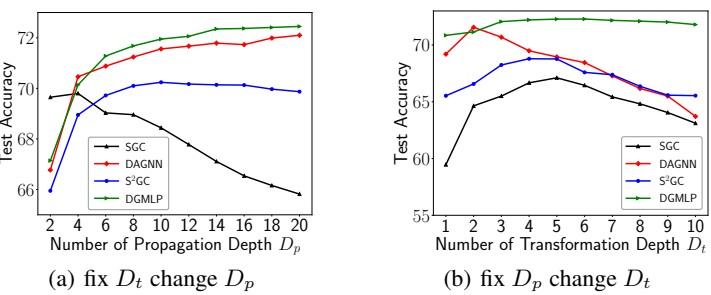

(a) fix $D_t$ change $D_p$         (b) fix $D_p$ change $D_t$

Figure 6: Test accuracy with different $D_p$ or $D_t$.

## 7.4 ANALYSIS OF MODEL DEPTH

In this subsection, we conduct experiments to validate that our proposed DGMLP can support both large $D_p$ and large $D_t$. We choose SGC, DAGNN, and S²GC as baseline methods.

Firstly, we fix $D_t$ to 3 and increase $D_p$ from 1 to 20 on the ogbn-arxiv dataset. As seen from Fig. 6(a), SGC cannot perform well when $D_p$ goes deep as the *over-smoothing* issue occurs. The predictive accuracy of DAGNN, S²GC, and our DGMLP maintains high when $D_p$ becomes large. Moreover, DGMLP consistently outperforms all the baseline methods when $D_p$ is greater than 6. The superiority of our DGMLP over DAGNN and S²GC lies in that we adopt a node-adaptive combination mechanism to satisfy the diverse needs of different nodes for propagation depth $D_p$.

Secondly, we fix $D_p$ to 10 and increase $D_t$ from 1 to 10. Fig. 6(b) shows that the predictive accuracy of all the baseline methods, including SGC, DAGNN, and S²GC decreases rapidly when $D_t$ becomes large. It is because that these methods do not take the *model degradation* issue into consideration, which is precisely the main contributor to the performance degradation when $D_t$ is large. This property limits the expressive power of these baseline methods, resulting in relatively low performance when adopted on large graphs. In the meantime, the performance of our proposed DGMLP still increases steadily or maintains even when $D_t$ is large. To sum up, compared with other baseline methods, our DGMLP can consistently improve predictive accuracy with larger $D_p$ or $D_t$, which validates our experimental analysis and guidelines in Sec. 4 and 5.

## 8 CONCLUSION

In this paper, we perform an experimental evaluation of current GNNs and find the root cause for the failure of deep GNNs: the *model degradation* issue introduced by large transformation depth. The *over-smoothing* issue introduced by large propagation depth does harm the predictive accuracy. However, GNN is much more sensitive to the *model degradation* issue than the *over-smoothing* issue, i.e., the *model degradation* issue happens much earlier than the *over-smoothing* issue as $D_p$ and $D_t$ increases at the same speed. Based on the above analysis, we present Deep Graph Multi-Layer Perceptron (DGMLP), a flexible and deep GNN model that simultaneously support large propagation and transformation depth. Extensive experiments on seven real-world graph datasets demonstrate that DGMLP outperforms state-of-the-art GNNs, and enjoys high scalability and efficiency at the same time.

## 9 REPRODUCIBILITY STATEMENT

The source code of DGMLP can be found in Anonymous Github (`https://anonymous.4open.science/r/DGMLP-4A79`). To ensure reproducibility, we have provided the overview of datasets and baselines in Section 7.1 and Appendix D.1. The detailed hyperparameter settings for our DGMLP can be found in Appendix D.2. Please refer to "README.md" in the Github repository for more reproduction details.

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

# A    DEEPER ANALYSIS OF GNN ARCHITECTURE

## A.1    CONVOLUTION PATTERN

According to whether the model disentangles the EP and ET operations, current GNNs can be classified into two major categories: entangled and disentangled, and each can be further classified into two smaller categories based on the order of EP and ET operations.

**Entangled Graph Convolution.**    Entangled Propagation and Transformation (**EPT**) pattern is widely adopted by mainstream GNNs, like GCN (Kipf & Welling, 2016), GraphSAGE (Hamilton et al., 2017), GAT (Velickovic et al., 2018), and GraphSAINT (Zeng et al., 2020b). The idea behind EPT-based GNNs is similar to the conventional convolution: it passes the input signals through a set of filters to propagate the information, which is further followed by nonlinear transformations. As a result, EP and ET operations are inherently intertwined in this pattern, i.e., each EP operation requires a neural layer to transform the hidden representations to generate the new embeddings for the next step. Motivated by ResNet (He et al., 2016), some GNNs with the **EPT-SC** pattern deepen the EPT-based GNNs with skip connections, and both JK-Net  (Xu et al., 2018), and ResGCN  (Li et al., 2019) are the representative methods of this category.

Besides the strict restriction that $D_p = D_t$, Table 2 shows EPT-based and EPT-SC-based GNNs also suffer from low scalability and low efficiency. On the one hand, a deeper structure has more parameters, resulting in greater computation costs. On the other hand, the number of nodes within each node's neighborhood grows exponentially with the increase of model depth in typical graphs, incurring significant memory requirement in a single machine or high communication costs in distributed environments (Zheng et al., 2020).

**Disentangled Graph Convolution.**   Previous works (Frasca et al., 2020; He et al., 2020; Wu et al., 2019; Zhang et al., 2021a) have shown that the true effectiveness of GNNs lies in the EP operation rather than the ET operation inside the graph convolution. Therefore, some disentangled GNNs propose to separate the ET operation from the EPT scheme. They can be classified into the following two categories according to the order of EP and ET operations.

One pattern is the Disentangled Transformation and Propagation (**DTP**). As shown in Table 2, DTP-based GNNs firstly execute the ET operations and then turn to the EP operations, and the most representative methods in this category are PPNP (Klicpera et al., 2018), APPNP (Klicpera et al., 2018), and AP-GCN (Spinelli et al., 2020). Unlike the DTP pattern, GNNs with the convolution pattern of Disentangled Propagation and Transformation (**DPT**) execute the EP operations in advance, and the propagated features are fed into a simple model composed of multiple ET operations (Zhang et al., 2021b). For example, SGC (Wu et al., 2019) removes the nonlinearities in GCN and collapses all the weight matrices between consecutive layers into a simple logistic regression. Based on SGC, SIGN (Frasca et al., 2020) further combines many graph convolutional filters in the EP operations, which differ in types and depths, utilizing significantly more structural information.

As shown in Table 2, compared with entangled GNNs, DTP-based GNNs can support large propagation depth $D_p$. Besides, DTP-based GNNs do not restrict $D_p = D_t$, enabling more flexibility in the GNN architecture design. However, they still have low scalability and efficiency issues when adapted to large graphs due to the high-cost propagation process during training. On the contrary, DPT-based GNNs only need to precompute the propagated features once. Therefore, they are easier to scale to large graphs with less computation cost and lower memory requirement.

## A.2    DGMLP VERSUS EXISTING METHODS

**EPT-based GNNs.** GNNs with the convolution pattern of EPT assume that $D_p = D_t$. However, as discussed in Sec. 6.1, the optimal $D_p$ and $D_t$ are highly related to graph sparsity and graph size. For example, it is better to assign large $D_p$ and small $D_t$ for better performance on a small graph with sparse edges. Due to the *model degradation* issue, current EPT-based GNNs usually have both small $D_p$ and $D_t$, owing to the inflexible restriction that $D_p = D_t$. Besides, the EPT-based GNNs also face the problem of low scalability and low efficiency. For example, GCN has the high time complexity of $\mathcal{O}(D_p Md + D_t Nd^2)$ for the need to repeatedly perform recursive neighborhood expansion at each training iteration to compute the hidden representations of each node. This process is unscalable due to the high memory and computation costs on a single machine and high communication costs in

Table 2: Algorithm analysis. We denote $N$, $N_l$, $M$ and $d$ as the number of nodes, the number of labeled nodes, edges and feature dimensions respectively. P is the EP operation and T is the ET operation. "SC" means "Skip Connection", and $k$ refers to the number of sampled nodes in GraphSAGE.

| Algorithm | Convolution Type | Pattern | Disentangled | Large $D_p$ | Large $D_t$ | Complexity |
|---|---|---|---|---|---|---|
| GCN (Kipf & Welling, 2016) | EPT | PT$\cdots$PT | $\times$ | $\times$ | $\times$ | $\mathcal{O}(D_p Md + D_t Nd^2)$ |
| GraphSAGE (Hamilton et al., 2017) | EPT | PT$\cdots$PT | $\times$ | $\times$ | $\times$ | $\mathcal{O}(k^{D_p} Nd^2)$ |
| JK-Net (Xu et al., 2018) | EPT-SC | PT$\cdots$PT | $\times$ | $\times$ | $\checkmark$ | $\mathcal{O}(D_p Md + D_t Nd^2)$ |
| ResGCN (Li et al., 2019) | EPT-SC | PT$\cdots$PT | $\times$ | $\times$ | $\checkmark$ | $\mathcal{O}(D_p Md + D_t Nd^2)$ |
| APPNP (Klicpera et al., 2018) | DTP | T$\cdots$TP$\cdots$P | $\checkmark$ | $\checkmark$ | $\times$ | $\mathcal{O}(D_p Md + D_t Nd^2)$ |
| AP-GCN (Spinelli et al., 2020) | DTP | T$\cdots$TP$\cdots$P | $\checkmark$ | $\checkmark$ | $\times$ | $\mathcal{O}(D_p Md + D_t Nd^2)$ |
| DAGNN (Liu et al., 2020) | DTP | T$\cdots$TP$\cdots$P | $\checkmark$ | $\checkmark$ | $\times$ | $\mathcal{O}(D_p Md + D_t Nd^2)$ |
| SGC (Wu et al., 2019) | DPT | P$\cdots$PT$\cdots$T | $\checkmark$ | $\checkmark$ | $\times$ | $\mathcal{O}(D_t N_l d^2)$ |
| SIGN (Frasca et al., 2020) | DPT | P$\cdots$PT$\cdots$T | $\checkmark$ | $\checkmark$ | $\times$ | $\mathcal{O}(D_t N_l d^2)$ |
| S$^2$GC (Rossi et al., 2020) | DPT | P$\cdots$PT$\cdots$T | $\checkmark$ | $\checkmark$ | $\times$ | $\mathcal{O}(D_t N_l d^2)$ |
| GBP (Chen et al., 2020b) | DPT | P$\cdots$PT$\cdots$T | $\checkmark$ | $\checkmark$ | $\times$ | $\mathcal{O}(D_t N_l d^2)$ |
| DGMLP | DPT | P$\cdots$PT$\cdots$T | $\checkmark$ | $\checkmark$ | $\checkmark$ | $\mathcal{O}(D_t N_l d^2)$ |

distributed environments. Compared with EPT-based GNNs, DGMLP is more flexible in assigning different $D_p$ and $D_t$ and enjoys better efficiency, scalability, and lower memory requirement.

**EPT-SC-based GNNs.** Compared with EPT-based GNNs, the skip connection in EPT-SC-based GNNs helps to alleviate the *model degradation* issue introduced by large $D_t$. However, Fig. 3(b) shows that the test accuracy degrades with deeper architecture due to the *over-smoothing* issue introduced by large $D_p$. A large graph without the graph sparsity issue requires large $D_t$ and small $D_p$. However, EPT-SC-based GNNs like ResGCN cannot satisfy such requirements since they simply restrict $D_p = D_t$. Besides, EPT-SC-based GNNs still have the issues of low scalability, efficiency, and high memory requirement as EPT-based GNNs. Compared with EPT-SC-based GNNs, DGMLP can improve the performance by enabling larger $D_p$ while maintaining high scalability, efficiency, and low memory requirement.

**DTP-based GNNs.** Compared with GNNs with the convolution pattern of EPT and EPT-SC, DTP-based GNNs disentangle the propagation and the transformation operation, thus can support flexible combinations of $D_p$ and $D_t$. As shown in Fig. 2(b) and Fig. 2(c), *over-smoothing* may have less impact on the performance degradation compared with *model degradation*, and the test accuracy will not decrease rapidly with the larger *over-smoothing* level. Therefore, lots of DTP-based GNNs own large $D_p$ and small $D_t$ and claim that they can perform well with the deeper architecture. However, they ignore the transformation depth $D_t$ in their design. As shown in Fig. 7(b), the test accuracy degrades rapidly if we increase the transformation depth $D_t$ of DAGNN. Correspondingly, it is hard for DTP-based GNNs to support large graphs with limited transformation depth $D_t$. Besides, as they execute ET operations first, scalability, efficiency, and memory requirement issues still exist.

**DPT-based GNNs.** Compared with current DPT-based GNNs, the proposed DGMLP utilizes the same convolution pattern. They all enjoy high scalability, efficiency, and low memory requirement as they can precompute the propagated features. The difference between DGMLP and other DPT-based GNNs is that DGMLP is the first method considering the smoothness from the node level. Therefore, DGMLP can assign individual feature propagation weights to different nodes and then get better-smoothed node embeddings by considering the personalized area in the graph where each node resides. Another advantage of DGMLP is that it can support large transformation depth $D_t$ with the help of residual connections.

# B  WHEN WE NEED DEEP GNN ARCHITECTURES?

**When we need deep EP?**  We review the related works and researches concerning $D_p$'s importance and then investigate experimentally when it is appropriate to enlarge $D_p$ in GNNs. GNNs mainly benefit from performing EP operations over neighborhoods. Stacking more EP operations will expand a node's receptive field and help it to gain more deep structural information. However, if the graph is dense, increasing $D_p$ may lead to *over-smoothing*, which results from the rapid expansion of the receptive field.

Table 3: Replacing the logistic regression in SGC with ResMLP. The test accuracy of the corresponding model on small graph Cora and large graph ogbn-arxiv when the MLP depth $D_t$ changes from 1 to 7.

| Dataset | 1 | 2 | 3 | 4 | 5 | 6 | 7 |
|---------|-----|------|------|------|------|------|------|
| Cora | 80.9 | **81.7** | 81.5 | 81.1 | 80.7 | 80.5 | 80.2 |
| ogbn-arxiv | 70.1 | 70.2 | 71.2 | 71.4 | 71.4 | **71.6** | 71.3 |

Table 4: Test accuracy under different edge sparsity.

| Datasets | $\frac{M}{N^2}$ | 2 | 4 | 6 | 8 | 12 | 16 | 20 |
|----------|------|------|------|------|------|------|------|------|
| Cora | 0.7‰ | **59.8** | 59.6 | 57.9 | 57.3 | 56.5 | 51.8 | 47.1 |
| PubMed | 0.1‰ | 78.5 | **78.9** | 77.8 | 77.6 | 77.3 | 76.6 | 75.8 |

Table 5: Test accuracy under different label missing rates.

| Datasets | Labels/class | 2 | 4 | 6 | 8 | 12 | 16 | 20 |
|----------|--------------|------|------|------|------|------|------|------|
| Cora | 20 | **81.5** | 81.3 | 80.8 | 80.1 | 80.0 | 79.5 | 78.8 |
|      | 1 | 53.8 | 59.2 | 62.9 | 64.3 | 66.5 | **66.7** | 65.6 |
| PubMed | 20 | 78.5 | **78.9** | 77.8 | 77.6 | 77.3 | 76.6 | 75.8 |
|        | 1 | 61.2 | 65.9 | 67.4 | 67.5 | 67.7 | **69.0** | 68.3 |

Generally, most real-world graphs exhibit sparsity in three aspects: edges, labels, and features. We define edge sparsity, label sparsity, and feature sparsity as follows: (1) Edge sparsity: nodes in real-world graphs usually have a skewed degree distribution, and many nodes are rarely connected (Kuramochi & Karypis, 2005). (2) Label sparsity: only a small part of nodes are labeled due to the high labeling costs or long labeling time (Garcia & Bruna, 2017). (3) Feature sparsity: some nodes in the graph do not own features, i.e., the new users or products in a graph-based recommendation system (Zhao & Akoglu, 2020). We argue that **sparse graphs naturally need deeper EP for larger receptive fields**. Experiments are conducted to demonstrate that the graph sparsity mentioned above will highly affect the optimal choice of $D_p$. SGC is adopted as the base model throughout these experiments.

Firstly, we investigate how edge sparsity influences the optimal $D_p$. For a fair comparison, we only sample part of the labels for Cora to have the same feature and label missing rates as PubMed. Then we increase $D_p$ from 2 to 20 on Cora and PubMed and report the test accuracy under each setting. The experimental results in Fig. 4 show that the optimal $D_p$ is 4 for PubMed and 2 for Cora, meaning that the sparser graph requires a larger $D_p$, which further demonstrate the benefits of deepening $D_p$ under edge sparse conditions. Secondly, we illustrate the relationship between the label sparsity and $D_p$. As shown in Fig. 5, we fix one labeled node per class on Cora and PubMed and keep the same setting with the edge sparsity experiment. It can be seen that the classification accuracy increases as $D_p$ ascends from 2 to 16. However, if we increase the label rate to 20 nodes per class, the node classification drops from 2 layers. The experiment on PubMed also shows that graphs with lower label rates require larger $D_p$. Finally, in Table 6, we report the test accuracy under the feature sparsity setting where 50% node features are randomly dropped from Cora and PubMed datasets. The result is consistent with the former two experiments that graphs with higher feature sparsity levels require larger $D_p$.

**When we need deep ET?** Generally, stacking multiple ET operations can better fit and learn the data distribution. However, we observe that the optimal $D_t$ for GNN is sensitive to the graph size. Concretely, **small graphs contain limited information, and shallow transformation architecture is enough for generating decent node embeddings. However, large graphs have complex structural information as complexity grows at a squared rate, which requires larger**

Table 6: Test accuracy under different feature missing rates.

| Datasets | Feature missing % | 2 | 4 | 6 | 8 | 12 | 16 | 20 |
|----------|-------------------|------|------|------|------|------|------|------|
| Cora | 0% | **81.5** | 81.3 | 80.8 | 80.1 | 80.0 | 79.5 | 78.8 |
| | 50% | 75.4 | 77.7 | 78.5 | **79.5** | 79.4 | 78.9 | 78.0 |
| PubMed | 0% | 78.5 | **78.9** | 77.8 | 77.6 | 77.3 | 76.6 | 75.8 |
| | 50% | 60.6 | 65.1 | 66.3 | 66.7 | 68.7 | **69.2** | 68.7 |

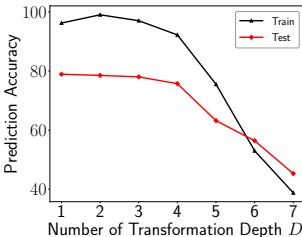 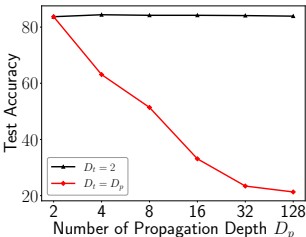 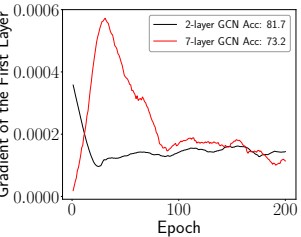

(a) Training and test accuracy both (b) Fixing $D_t = D_p$ degrades (c) First layer gradient comparison drop sharply when model grows the performance badly when $D_p$ of GCN with different layers on deep. becomes large on Cora dataset. Cora dataset.

Figure 7: Overfitting, entanglement, and gradient vanishing are not the major cause for the performance degradation in deep GNNs.

$D_t$ **to extract meaningful information.** We prove this insight by experimentally examining how $D_t$ influences the node classification performance on Cora and ogbn-arxiv datasets. Concretely, we replace the logistic regression in SGC with "MLP+Res" and increase $D_t$ from 1 to 7. Table 3 shows that the accuracy on ogbn-arxiv increases as $D_t$ ascends from 1 to 6. However, $D_t = 2$ is enough for the small graph Cora. Therefore, larger graphs need larger $D_t$ for high-quality node embeddings.

## C  MORE MISLEADING EXPLANATIONS

### C.1  OVER-FITTING

Some works (Rong et al., 2019; Li et al., 2019; Zhou et al., 2020b; Yang et al., 2020) attribute the performance degradation of deep GNNs to over-fitting. Concretely, over-fitting comes from the case when an over-parametric model tries to fit a distribution with limited training data, which results in learning well on the training data but failing to generalize to the testing data. We plot the corresponding node classification accuracy of GCN on both the training and the test set under different model depths in Fig. 7(a).

**Finding.**   **Both the training and test accuracy drop quickly when the model grows deep.** However, over-fitting assumes that over-parametric models get low training error but high testing error, inconsistent with the experimental results.

**Implication.** Over-fitting is not the primary cause for performance degradation of deep GNNs.

### C.2  ENTANGLEMENT

Some recent works (He et al., 2020; Liu et al., 2020) argue that the key factor compromising the performance of deep GNNs is the entanglement of EP and ET operations in current graph convolutional layers. For example, DAGNN claims that the entanglement of EP and ET operations in GCN is the true reason it only supports shallow architectures. To investigate this statement, we vary the transformation depth $D_t$ in ResGCN, DenseGCN, and vanilla GCN and report their test accuracy on the PubMed dataset. The experimental results are shown in Fig. 3(b).

**Finding 1.** **Although ResGCN and DenseGCN have entangled designs, when $D_p$ and $D_t$ both become large, they do not experience significant performance drop as GCN does.** Both the residual and dense connections can effectively alleviate the influence of *model degradation*, and the performance degradation of ResGCN and DenseGCN starting at $D_t = 6$ might be caused by the *over-smoothing* issue.

**Implication.** GNNs can go deep even in an entangled design, and the entanglement of EP and ET operations may not be the true limitation of the GNN depth.

What is worth noting is that previous works (Zhu & Koniusz, 2021; Liu et al., 2020), which have disentangled designs and state that they support deep architectures, are only able to go deep on the propagation depth $D_p$. In their original design, if we increase $D_t$, their performance will also degrade badly. To validate this, we run DAGNN in two different settings: the first controls $D_t = 2$ and increases $D_p$, the second controls $D_t = D_p$ and increases $D_p$. The test accuracy under these two settings on the PubMed dataset is shown in Fig. 7(b).

**Finding 2.** **The disentangled DAGNN also experiences huge performance degradation when the model owns large $D_t$.** If the entanglement dominates the performance degradation of deep GNNs, DAGNN should be able to also go deep on $D_t$. However, if we individually increase $D_t$ of DAGNN, the sharp performance decline still exists.

**Implication.** The major limitation of deep GNNs is the *model degradation* issue introduced by large $D_t$ rather than the entanglement of EP and ET operation.

### C.3 GRADIENT VANISHING/EXPLOSION

Gradient vanishing means that the low gradient in the shallow layers makes it hard to train the model weights when the network goes deeper, and it has a domino effect on all of the further weights throughout the network. To evaluate whether the gradient vanishing exists in deep GNNs, we respectively perform node classification experiments on the Cora dataset and plot the gradient – the mean absolute value of the gradient matrix of the first layer in the 2-layer and 7-layer GCN in Fig. 7(c).

**Finding 1.** **Although the test accuracy of 7-layer GCN drops quickly, its gradient is as large as the gradient of 2-layer GCNs, or even larger in the initial model training phases.** The explanation for the initial gradient rise of the 7-layer GCN might be that the large model needs more momentum first to adjust and then jump out of the suboptimal local minima initially.

**Implication.** Gradient vanishing is not the leading cause of performance degradation in deep GNNs.

### C.4 DISENTANGLED CONVOLUTION

We then investigate **why disentangling EP and ET is able to allow more EP operations.** Concretely, we carefully investigate current disentangled GNNs (Frasca et al., 2020; Rossi et al., 2020) and find that the decoupling strategy makes the propagation and the transformation operations independent, thus $D_p$ and $D_t$ are not forced to be the same. Therefore, disentangled GNNs generally fix $D_t$ and increase $D_p$ to capture deeper graph structural information. Here we select two disentangled GNNs, $S^2GC$ and Grand, which state that they support deep architectures. Their performance of individually increasing $D_p$ and $D_t$ are shown in Fig. **??** and Fig. **??**, respectively.

**Finding.** **Individually increasing $D_p$ would not incur a severe performance drop even when $D_p$ is increased to 64, while their performance would experience a sharp decline when $D_t$ increases.** Increasing $D_p$ enlarges each node's receptive field, thus leading to more decent representations, which is demonstrated to be even more beneficial for sparse graphs in our following experiments (See Sec. 6.1).

**Implication.** Deep disentangled GNNs are flexible to individually increase $D_p$, so that the *model degradation* introduced by large $D_t$ can be avoided.

Table 7: Overview of datasets and task types.

| Dataset | #Nodes | #Features | #Edges | #Classes | #Train/Val/Test | Description |
|---|---|---|---|---|---|---|
| Cora | 2,708 | 1,433 | 5,429 | 7 | 140/500/1,000 | citation network |
| Citeseer | 3,327 | 3,703 | 4,732 | 6 | 120/500/1,000 | citation network |
| Pubmed | 19,717 | 500 | 44,338 | 3 | 60/500/1,000 | citation network |
| ogbn-arxiv | 169,343 | 128 | 1,166,243 | 40 | 91K/30K/47K | citation network |
| ogbn-products | 2,449,029 | 100 | 61,859,140 | 47 | 196K/49K/2,204K | citation network |
| ogbn-papers100M | 111,059,956 | 128 | 1,615,685,872 | 172 | 1,207K/125K/214K | citation network |
| Industry | 1,000,000 | 64 | 1,434,382 | 253 | 5K/10K/30K | short-form video network |

Figure 8: Performance along with training time on the Industry dataset.

# D  EXPERIMENTAL DETAILS

## D.1  DATASET DETAILS

The details of the adopted seven datasets can be found in Table 7.

## D.2  HYPERPARAMETER DETAILS

For the three small citation networks, we adopt a simple logistic regression as the classifier. The propagation depth $D_p$ is set to 20 for Cora and PubMed, and 15 for Citeseer. The learning rate is set to 0.1, and the dropout rate is obtained from a search of range 0.1 to 0.5 with step 0.1. Residual connections are not used on these three citation networks.

For three large ogbn datasets and Industry dataset, a 6-layer MLP with hidden size of 512 is used, and the propagation depth $D_p$ is respectively 20 and 12 for them. Residual connections are used for these four large datasets. Besides, the temperature $T$ is set to 1 as default if it is not specified, and the learning rate and the dropout rate are set to 0.001 and 0.5, respectively.

We run all the methods for 200 and 500 epochs on the citation networks and OGB datasets, respectively. Besides, we run all methods 10 times and report the mean values and the variances of different performance metrics. Other hyperparameters are tuned with the toolkit OpenBox (Li et al., 2021) or follow the settings in their original paper.

# E  ADDITIONAL EXPERIMENTS

## E.1  PERFORMANCE-EFFICIENCY ANALYSIS

In this subsection, we evaluate the efficiency of each method in our industrial environment: the real-world Industry dataset. Here, we precompute the smoothed features of each DPT-based GNN, and the time for preprocessing is also included in the training time. Fig. 8 illustrates the results on the Industry dataset across representative baseline methods of each convolution pattern.

Compared with DPT-based GNNs, we observe that both EPT-based and DTP-based GNNs require a significantly larger training time. For example, GCN takes 32 times longer than SGC to complete the training and the training time of AP-GCN is 112 times the one of SGC. Due to the more complex preprocessing, our DGMLP takes twice the training time of SGC's. However, the relatively time-consuming preprocessing brings significant performance gain to DGMLP, exceeding SGC by

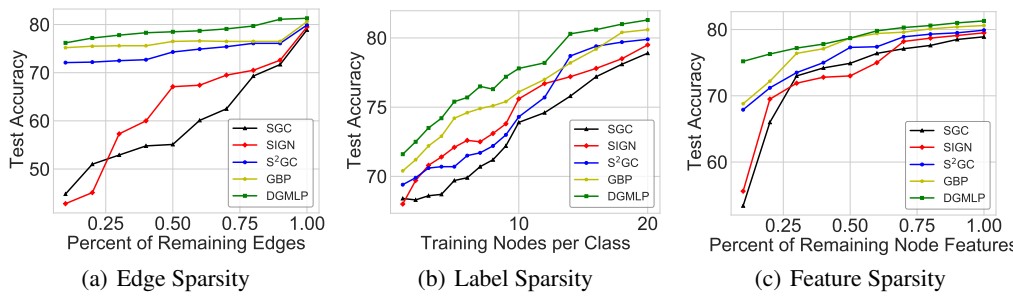

| (a) Edge Sparsity | (b) Label Sparsity | (c) Feature Sparsity |

Figure 9: Test accuracy on PubMed dataset under different levels of feature, edge and label sparsity.

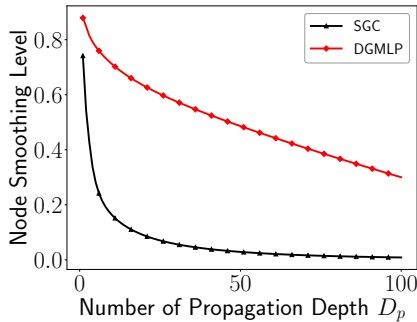

Figure 10: Graph Smoothing Level comparison between SGC and our proposed DGMLP.

more than 2% across three citation networks. To sum up, our proposed DGMLP achieves better performance and maintains high efficiency.

### E.2 INFLUENCE OF GRAPH SPARSITY

To simulate extreme sparse situations in the real world, we design three independent settings on the PubMed dataset to test the performance of our proposed DGMLP when faced with edge sparsity, label sparsity, and feature sparsity, respectively.

**Edge Sparsity.** We randomly remove some edges in the original graph to strengthen the edge sparsity situation. The edges removed from the original graph are fixed across all the methods under the same edge remaining rate. From the results in Fig. 9(a), it is clear to see that the performance of GBP, S$^2$GC, and our DGMLP is significantly better than SIGN and SGC. Further, DGMLP always has higher test accuracy than GBP and S$^2$GC. SIGN and SGC both have a flaw in that they can not effectively capture the deep structural information, which would become more prominent when edges are extremely sparse in the graph.

**Label Sparsity.** In this setting, we vary the training nodes per class from 1 to 20 and report the test accuracy of each method. The experimental results in Fig. 9(b) show that the test accuracy of all the compared methods increases as the number of training nodes per class becomes larger. In the meantime, our DGMLP outperforms all the baselines throughout the experiment. With 11 training nodes per class, DGMLP achieves comparable performance with SGC trained under 20 training nodes per class.

**Feature Sparsity.** In a real-world situation, the feature of some nodes in the graph might be missing. We follow the same experimental design in the edge sparsity setting but removing node features instead of edges. The results in Fig. 9(c) illustrate that our proposed DGMLP has a great anti-interference ability when faced with feature sparsity as its performance drops only a little even there is only 10% node features available.

### E.3 INTERPRETABILITY

In this subsection, we empirically explain why our proposed DGMLP is more robust to the *over-smoothing* issue. Concretely, the baseline method SGC is used for comparison, and the temperature $T$ in our DGMLP is set to 0.2. Besides, the Graph Smoothing Level ($GSL$) in Sec. 3.2.1 is adopted to evaluate the graph smoothness.

The results in Fig. 10 show that the $GSL$ of both SGC and DGMLP decreases as $D_p$ increases. However, the descending speed of SGC is much quicker than DGMLP, especially when $D_p$ changes from 1 to 5. Moreover, the $GSL$ of DGMLP at $D_p = 100$ is even larger than the one of SGC at $D_p = 5$. Fig. 10 strongly illustrates that our proposed DGMLP is more robust to the *over-smoothing* issue introduced by large $D_p$, and the performance results in Fig. 6(a) further shows that DGMLP can take advantage of this property to gain more beneficial deep structural information for prediction.

