# OpenReview forum: "Evaluating Deep Graph Neural Networks"
_ICLR.cc/2022/Conference — ICLR 2022 Submitted_

### Official Review · Reviewer_tpnS · 2021-10-29

**Correctness:** 2
**Technical Novelty And Significance:** 2
**Empirical Novelty And Significance:** 2
**Recommendation:** 3
**Confidence:** 4

**Main Review:**

The definitions of Node Smoothing Level and Graph Smoothing Level compare the representations of the same node across different EP steps. However, the oversmoothing problem is across nodes. Considering the deviations, the performance of DGMLP appears to be comparable to the state-of-the-art. It is claimed that DGMLP can support large transformation depth D_t. But Fig6 (b) only shows a very stable accuracy as D_t increases. It would be more convincing if the performance of DGMLP improves with a larger D_t. Otherwise, why bother to increase D_t? Larger D_t means more parameters, and hence may need to train models longer (and with better training settings). Is it possible that large D_t leads to undertrain of GNN? The proposed node-adaptive weighting mechanism is interesting. It could helpful to see experiments that validate the importance of the node-adaptive weighting mechanism.

**Summary Of The Paper:**

In this work, the authors performed an experimental evaluation on several GNNs in order to understand what aspects of the current architecture designs that leads to the compromised performance of deep GNNs. They claimed to find the root causes: large propagation depth leads to the over-smoothing issue and large transformation depth leads to the model degradation issue. Then they proposed guidelines for designing deep GNNs and a Deep Graph Multi-Layer Perceptron (DGMLP) for implementing the guidelines. Experimental results demonstrate performance of DGMLP in accuracy, flexibility, scalability, and efficiency.

**Summary Of The Review:**

The experiments, which compare various GNNs, provide useful information. The proposed DGMLP contains interesting ideas. However, experimental results are mixed and do not clearly support all claims made in this manuscript. The proposed smoothing level metrics do not measure smoothing across nodes.

---

> ### Author Response · Authors · 2021-11-24
> **Response to Reviewer-4**
>
> We thank the reviewer for your insightful reviews and valuable time. We address all concerns as follows.
>
> ### 1. The definitions of Node Smoothing Level
> Similar to Instance Information Gain proposed in [1], we consider the similarity between the input feature information and final representation in Eq.(5). Besides, we further propose to consider the similarity between the final representation and the representation at the stationary (over-smoothed) state. A smaller distance to the stationary state means the node feature is more likely to be over-smoothed. We agree that it is more obvious if we directly measure the similarity across nodes, but our proposed Node Smoothing Level is also a reasonable metric for the over-smoothing issue. Besides, as this definition compares the representations of the same node across different EP steps, it is easier to guide the node-adaptive weighting mechanism proposed in our DGMLP model.
>
>
> ###  2. The performance of DGMLP improves with a larger $D_t$
> Figure 6(b) shows a steady performance increase when $D_t$ changes from 1 to 4, and then the predictive accuracy holds as $D_t$ increases. As shown in Table 3 in Appendix B, we replace the logistic regression in SGC with "MLP+Res" and increase $D_t$ from 1 to 7 on the ogbn-arxiv dataset. Table 3 shows that the predictive accuracy on the ogbn-arxiv dataset steadily increases as $D_t$ ascends from 1 to 6. However, $D_t = 2$ is enough for the small graph Cora. Therefore, larger graphs need larger $D_t$ for high-quality node embeddings. We will add the experiments on larger graphs (i.e., ogbn-papers100M) to better verify the necessity of the large $D_t$.
>
> ###  3. The importance of the node-adaptive weighting mechanism.
> The key difference between DGMLP and other baselines (e.g., SGC and S$^2$GC) in Figure 6 is its node-adaptive weighting mechanism. As shown in Figure 6(a), we fix $D_t$ and increase $D_p$, and report the corresponding test accuracy on the ogbn-arxiv dataset when the propagation depth $D_p$ changes from 2 to 20. The experimental results show DGMLP consistently outperforms all the baseline methods when $D_p$ is greater than 6, which verifies the effectiveness of our node-adaptive weighting mechanism.
>
> [1] Zhou, Kaixiong, et al. "Towards Deeper Graph Neural Networks with Differentiable Group Normalization." NeurIPS 2020.
>
> We are very glad to respond if you have any new questions.
>
> Respectfully,
>
> Paper753 Authors

---

### Official Review · Reviewer_ddQY · 2021-11-01

**Correctness:** 2
**Technical Novelty And Significance:** 1
**Empirical Novelty And Significance:** 1
**Recommendation:** 3
**Confidence:** 5

**Details Of Ethics Concerns:**

Not applicable.

**Main Review:**

**Strengths:**

(1) The concepts of embedding propagation and transformation depths are formally defined to study their impacts on deep GNNs.
(2) The series of empirical studies on these two depths are given.
(3) The paper is well organized.


**Weakness:**

(1) The empirical guidelines 1-3, and the experimental analysis (in Section 4) are not novel at all to the community of deep GNNs. Specifically, the decoupling of embedding propagation and transformation, adaptive weighting, and skip connection have already been studied in literature. I know the authors derive these guidelines based on formulating the concepts of D_p & D_t and conducting the extensive experiments to get Findings 1. However, these knowledges have been implicitly explained in prior efforts. For example, the decoupling of EP and ET was proposed in DAGNN [1], based on the empirical study of SGC similar to the one in this paper. To avoid the harmful impact of embedding transformation, the identity mapping is incorporated in GCNII [4] to avoid the overly feature change. In other word, the dominant influence of embedding transformation in deep GNNs is not a news.

(2) The proposed model, named DGMLP, is incremental comparing with the existing works, including DAGNN, SGC, JK-Net, etc. The only difference is the usage of skip connection in the final MLP module.

(3) Although the experimental results show the superiority of DGMLP, most of the important baselines are missing in Table 1, which easily leads to misunderstanding. For example, GCNII has good or comparable performances in Cora, Citeseer, and ogbn-arxiv. In obgn-products, SAGN [5] and DeeperGCN [6] with the similar architecture to the proposed model shows outperforming results. Furthermore, SAGN also has better results in ogbn-100M. But all these important baselines are missing.

(4) Some of the claims made in this paper is not correct. See below for the detailed comments.


**Questions:**
(1) Under Eq. (4), it is stated that ``Under this scenario, the neighborhood information is fully corrupted, resulting in catastrophic node classification accuracy.” I’m concerning on the correctness of this statement: why neighborhood information will be corrupted if the neighborhood influence is determined by the degree? The specific neighbor features could still distinguish them. As shown in Figure 4 of reference [1], by stacking many EP layers, the node classification accuracy is still good enough, which empirically demonstrates that neighbor information is not corrupted.

(2) The claim of ``As the over-smoothing issue is only introduced by the EP operation rather than the ET operation” below Eq. (4) is not correct. As theoretically demonstrated in references [2,3], the over-smoothing issue is also correlated to the singular values of trainable weights and non-linear activations involved in ET.

(3) For the example study of Figure 2(c), it is unclear how to formulate GCN model with D_t=2 and the adjacency matrix powered by D_p/2. Could you illustrate more with the formal expression or model figure?

(4) In Section 4.2, please formally define the residual connection and dense connection used in the empirical study.

(5) The guideline 2 seems not novel at all. It is well-known that the optimal convolution depths for the diverse nodes are different. For example, DAGNN in [1] applies attention module to learn the optimal combination weight for each specific node.

[1] Liu, Meng, Hongyang Gao, and Shuiwang Ji. "Towards deeper graph neural networks." Proceedings of the 26th ACM SIGKDD International Conference on Knowledge Discovery & Data Mining. 2020.

[2] Cai, Chen, and Yusu Wang. "A note on over-smoothing for graph neural networks." arXiv preprint arXiv:2006.13318 (2020).

[3] Oono, Kenta, and Taiji Suzuki. "Graph neural networks exponentially lose expressive power for node classification." arXiv preprint arXiv:1905.10947 (2019).

[4] Chen, Ming, et al. "Simple and deep graph convolutional networks." International Conference on Machine Learning. PMLR, 2020.

[5] Sun, Chuxiong, and Guoshi Wu. "Scalable and Adaptive Graph Neural Networks with Self-Label-Enhanced training." arXiv preprint arXiv:2104.09376 (2021).

[6] Li, Guohao, et al. "Deepergcn: All you need to train deeper gcns." arXiv preprint arXiv:2006.07739 (2020).

**Summary Of The Paper:**

This paper points out the dominant influence on deep GNNs, i.e., the embedding transformation, with careful and extensive studies. Based on these studies, several guidelines are proposed to design deep model named DGMLP.

**Summary Of The Review:**

This paper formally define the concepts of embedding propagation and transformation depths, and distinguish their impacts on the deep GNNs. Based on the empirical studies, some well-known guidelines are proposed to design deep GNNs.

However, I cannot find much novelty from this paper, which seems to ensemble the existing knowledges. Some of important baselines are missing to validate the effectiveness of the proposed methods.

---

> ### Author Response · Authors · 2021-11-24
> **Response to Reviewer-3, Part 1**
>
> We thank the reviewer for your insightful reviews and valuable time. We address all concerns as follows.
>
> ### 1. The empirical guidelines
> We agree that the disentanglement of embedding propagation and transformation, adaptive weighting, and skip connection have already been studied in the previous literature. However, the true reason why GNN cannot go deep has not been well understood in previous work.
>
>
> For example, although the decoupling of EP and ET is proposed in DAGNN, this work claims that the entanglement of EP and ET operations is the main bottleneck of deep GNNs. As introduced in Appendix C.2 in our original manuscript, GNNs can go deep even in an entangled design (e.g., ResGCN), and the entanglement of EP and ET operations may not be the true limitation of the GNN depth.
> As we introduced in Section 5.3, GCNII addresses large $D_t$ via initial residual connections and identity mappings. However, as stated in this paper that "it remains an open problem to design a GCN model that effectively prevents over-smoothing and achieves state-of-the-art results with truly deep network structures." Although the design of GCNII can also tackle the model degradation problem in deep embedding transformation, it still aims to solve the over-smoothing problem in the original paper. To sum up, the model degradation issue introduced by large $D_t$ in GNNs has not been well understood in previous work.
>
>
> ### 2. The novelty of DGMLP
> Our node-adaptive weighting mechanism is also new in the GNN architecture design. To the best of our knowledge, DGMLP is the first attempt to consider the node-wise feature propagation in a heuristic and non-parameter manner in scalable GNNs.
>
> Details: 1) Although SGC is scalable, it only considers the propagated features at a single fixed hop. Instead, we combine multi-scale propagated features in a node adaptive manner to make better use of the graph information. 2) Although DAGNN is also node-adaptive, it may face the scalability issue when applied on large graphs. In contrast, the embedding propagation of SGC-based methods (e.g., GBP [1], SIGN [2], and DGMLP) are executed before training, which avoids performing recursive embedding propagation at each training epoch and storing the entire adjacency matrix on GPU. In this way, our method is more scalable and efficient to apply to large graphs. 3) JK-Net is an entangled model, and it may face the scalability issue and over-smoothing issue when exploring a large propagation depth $D_p$. The only similarity between DGMLP and JK-Net is the use of skip connections.
>
> ### 3. Missing baselines
> We propose the node-adaptive weighting mechanism to address the over-smoothing problem and the skip-connection to address the model degradation problem. To better understand where the main performance gain comes from and the correctness of our findings, we seek a simple but effective design principle in DGMLP.
> The attention mechanism for node adaptive propagation is not adopted, and we just use a more interpretable and heuristic weighting mechanism in DGMLP. For example, we can replace this mechanism with the attention mechanism proposed in SAGN [3] and get better performance. To sum up, the main contribution of the proposed DGMLP is not to outperform the current state-of-the-art baselines but to verify the correctness of our findings with respect to the over-smoothing and model degradation problems.
>
> ### 4. Fully corrupted neighborhood information
> According to Equation (4), after infinite times of multiplication, the influence from node $v_i$ to $v_j$ is only determined by their degrees. Suppose $\small \mathbf{X}^{(0)}=\mathbf{X}$ is the original node feature matrix, and $\small \mathbf{X}^{(k)}=\mathbf{\hat{A}}^{k}\mathbf{X}^{(0)}$ is the smoothed features after $k$ times of EP operation. If we set $r = 0.5$ and get $\small \mathbf{\hat{A}} = \widetilde{\mathbf{D}}^{\frac{1}{2}}\widetilde{\mathbf{A}}\widetilde{\mathbf{D}}^{-\frac{1}{2}}$, node $v_i$ and $v_j$ will get the same propagated node embeddings if their degree is same. Besides, if we set $r = 0$ and get $\small \mathbf{\hat{A}} = \widetilde{\mathbf{D}}^{-1}\widetilde{\mathbf{A}}$, the propagated feature of all nodes will be the same. In fact, such analysis has also been recognized in the paper of S$^2$GC.
>
> From Figure 4 of DAGNN[4], it is true that the node classification accuracy is still good enough by stacking 50 EP layers. However, as we still increase the number of EP layers to 500, the test accuracy is extremely low, which verifies the conclusion in Equation (4) since it assumes an infinite number of EP layers.
>
> ### 5. The claim of the over-smoothing
> We agree that "the over-smoothing issue is only introduced by the EP operation rather than the ET operation" is too aggressive. We will change the claim that the over-smoothing issue is mainly introduced by the EP operation rather than the ET operation.

---

> > ### Author Response · Authors · 2021-11-24
> > **Response to Reviewer-3, Part 2**
> >
> > ### 6.  Example study of Figure 2\(c\)
> > For example, to get the model with $\small D_t = 2$ and $\small D_p = 10$ in Figure 2\(c\), we only need to set the normalized matrix $\small \mathbf{\hat{A}}$ to $\small \mathbf{\hat{A}} = \mathbf{\hat{A}}^{(D_p/2)} = \mathbf{\hat{A}}^{(10/2)} = \mathbf{\hat{A}}^{(5)}$.
> >
> > ### 7. Formally define the residual connection and dense connection
> >
> > Thanks for the advice. The residual and dense connections are not formally defined since they are the same as the paper[5]. We will add the definition in the revised manuscript to make it clearer.
> >
> >
> >
> > ### 8. The novelty of guideline 2
> > We agree that DAGNN has previously proposed this guideline. Considering the scalability issue of GNN, our method follows the SGC-based paradigm, and guideline 2 is directly motivated by the experimental results of SGC in Figure (4). Although DAGNN is node-adaptive, it cannot be easily scaled to large-scale graphs since it executes embedding propagation during training.
> >
> > [1] Chen, Ming, et al. "Scalable Graph Neural Networks via Bidirectional Propagation." NeurIPS 2020.
> >
> > [2] Frasca, Fabrizio, et al. "SIGN: Scalable Inception Graph Neural Networks." arXiv 2020.
> >
> > [3]  Sun, Chuxiong, and Guoshi Wu. "Scalable and Adaptive Graph Neural Networks with Self-Label-Enhanced training." arXiv preprint arXiv:2104.09376 (2021).
> >
> > [4] Liu, Meng, Hongyang Gao, and Shuiwang Ji. "Towards deeper graph neural networks." Proceedings of the 26th ACM SIGKDD International Conference on Knowledge Discovery & Data Mining. 2020.
> >
> > [5] Li, Guohao, et al. "Deepgcns: Can gcns go as deep as cnns?." ICCV 2019.
> >
> > We are very glad to respond if you have any new questions.
> >
> > Respectfully,
> >
> > Paper753 Authors

---

### Official Review · Reviewer_QtrD · 2021-11-02

**Correctness:** 3
**Technical Novelty And Significance:** 3
**Empirical Novelty And Significance:** 3
**Recommendation:** 6
**Confidence:** 4

**Main Review:**

The exploration of hyper-parameters for this problem is well founded, especially given the many different architectures that have been proposed.  The paper characterizes this through two primary parameters, Dp and Dt, and compares the DGMLP with varying values.  It isn’t clear why the authors don’t refer to these as hyper-parameters (except buried in the appendix). Perhaps it is not surprising that various papers have shown that the unentangled approach can outperform the entangled one.

The main body of the paper seems repetitive at times, stating repeatedly the basic points about the tradeoffs with Dp and Dt.  In the reviewers opinion, the meaningful portions of the paper are in the appendices, especially A and B.  It isn’t clear why these don’t make up the major portion of the main paper.  It seems that the paper spends a large amount of space with discussion and generality, often spending too much effort on criticism.

The results show run-time and GPU memory versus graph size, and the DGMLP has good scaling properties.

The node-adaptive weighting mechanism (Section 6.1) is interesting.  However, the influence here wasn’t entirely clear.

The paper would be significantly improved with a statement of the DGMLP in some kind of Table or Algorithm statement.

The DGMLP is able to achieve state of the art results compared with other algorithms.  So, the value here is the flexibility of the DGMLP, and also perhaps complexity reduction and scalability. It wasn’t entirely clear what is the overall complexity in comparison with other algorithms, especially with respect to choice of the hyper-parameters.  Perhaps the authors could add some further complexity results that include the cost of hyper-parameter exploration with the various algorithms to make this clear to the reader.


**Summary Of The Paper:**

The paper considers the application of GNNs to semi-supervised node classification tasks. The problem arises in big data problems such as modeling citation databases.  In this line of work it is assumed that the data follows a graph structure, i.e., that locally connected nodes are likely to have the same label. Each node has a d-dimensional feature vector, with graph degree matrix A. Various GNN architectures have been proposed and studied for this problem, often using citation databases as the driving application. The paper considers two aspects of this problem.  First, the authors consider various choices of the hyper-parameters, including the GNN iterations and a (typically) fully connected layer at the output.  It is well known that iterating the graph convolutions leads to local averaging that will asymptotically converge to some form of smoothing, which can result in poor classification.  Here, the authors decouple the GNN smoothing and the follow on network, and refer to this as “unentangled” processing architecture (also called decoupled in the numerical results portion).  They further propose a revised GNN architecture (DGMLP) that incorporates node-adaptive weighting and skip-connections in the second stage.  Numerical experiments complete the work, showing that with appropriate choice of parameters the DGMLP is flexible and can achieve the state of the art results, and good scalability.



**Summary Of The Review:**

The theme of the paper is “Hyper-Parameter Selection and a Modified GNN for Semi-Supervised Node Classification”.  Perhaps that’s a better title.

Overall the paper builds on the many other GNN works for this problem, incorporates the best known practices, and provides a flexible approach to achieving the state of the art results when compared to other algorithms.  However, the presentation could be significantly strengthened as outlined in the Main Review.

---

> ### Author Response · Authors · 2021-11-24
> **Response to Reviewer-2**
>
> Thanks for your insightful feedback. We appreciate your assessment about this paper that "The exploration of hyper-parameters for this problem is well founded, especially given the many different architectures that have been proposed." The answers to your concerns are as follows.
>
> ### 1. Meaningful portions of the paper
> We agree that Appendix A and Appendix B are meaningful portions of this paper. However, due to the page limit, we have to put these two parts in the Appendix since the most important contribution of this paper is to find the true reason why most GNNs cannot go deep.
>
> ### 2. The influence of node-adaptive weighting
> The key difference between DGMLP and other baselines (e.g., SGC and S$^2$GC) in Figure 6 is its node-adaptive weighting mechanism. As shown in Figure 6(a), we fix $D_t$, and report the corresponding test accuracy on the ogbn-arxiv dataset when the propagation depth $D_p$ changes from 2 to 20. The experimental results show DGMLP consistently outperforms all the baseline methods when $D_p$ is greater than 6, which verifies the effectiveness of our node-adaptive weighting mechanism.
>
> ### 3. Statement of the DGMLP
> Thanks for your advice! We will introduce DGMLP with a Table or Algorithm statement in the revised manuscript.
>
> ### 4. Overall complexity in comparison with other algorithms
> Thanks for mentioning the complexity analysis. We have compared the complexity of DGMLP with other baselines in Table 2 of Appendix A.2.
>
> ### 5. The theme of the paper
> We agree that this title may not be appropriate, and we will improve it in the revised manuscript.
>
> We are very glad to respond if you have any new questions.
>
> Respectfully,
>
> Paper753 Authors

---

> > ### Comment · Reviewer_QtrD · 2021-11-29
> > **Reply to the Authors response**
> >
> > Thank you for your reply to the initial review.
> >
> > 1. I continue to believe that the writing style can be improved, and that important technical portions of the Appendix A and B can and should be incorporated into the body of the paper.  It still seems that the paper explores hyper parameters, and also adds node adaptivity.  These build on the other prior GNN works (many of high quality), and your methods seem like natural next steps, rather than some dramatically new interpretation that other authors have somehow missed.
> >
> > 2. I know that the node adaptivity is part of the algorithm, but still wondering if there is a way to better explore the value of this, e.g., can you add it into other existing algorithms?  Is this another modification that can be used generally?
> >
> > 4. Complexity:  Table 2 shows "order of" type complexity, which is very useful.  However, because basic claims of the paper are that the proposed algorithm is preferred due to its generality and scaling (and not that it necessarily achieves better performance), then some detailed comparison is important.  This should include the need for selecting the hyper-parameters.

---

> > > ### Author Response · Authors · 2021-11-30
> > > **Response to Reviewer-2**
> > >
> > > Thanks for your continuous comments! We address your concerns as follows.
> > >
> > > ### 1. Writting style
> > > We agree that Appendix A and Appendix B can and should be incorporated into the body of the paper.
> > > Specifically, we summarized the current deep GNN architectures in Appendix A and analyzed when we need deep GNN architecture in Appendix B. These two parts are meaningful to this work, and we will incorporate these two parts into the main body in the revised manuscript.
> > >
> > > ### 2. A better way to explore the value of node adaptivity
> > > This advice is constructive! In fact, our node adaptivity can be easily integrated into other existing GNN algorithms. In this work, we only consider introducing the node adaptivity to DPT-based GNNs(i.e., SGC, SIGN, and GBP) since 1) they don't restrict $D_p = D_t$; 2) they are more scalable than other GNN methods while achieving comparable or even better node classification performance. However, our node adaptivity mechanism can be used more generally, and we can introduce it to EPT-based GNNs, EPT-SC-based GNNs, and DTP-based GNNs with little modification.
> > >
> > > ### 3. Detailed comparison
> > > The aim of GAMLP is not to outperform the current state-of-the-art baselines but to verify the correctness of our findings with respect to the over-smoothing and model degradation problems. We agree that we should add a more detailed comparison, including selecting the hyper-parameters. We will update the comparison in detail in the revised manuscript.
> > >
> > > Since the discussion period will be closed on November 29, we are very happy to respond if you have additional comments or concerns on our responses.
> > >
> > > Respectfully,
> > >
> > > Paper753 Authors

---

### Official Review · Reviewer_4Jnf · 2021-11-03

**Correctness:** 2
**Technical Novelty And Significance:** 2
**Empirical Novelty And Significance:** 3
**Recommendation:** 5
**Confidence:** 5

**Main Review:**

(+) The paper is well-organized and easy to follow.

(+) This work performs a systematic study to analyze the main issues of the difficulty in training deep GNNs by disentangling the effects of embedding propagation (EP) and embedding transformation (ET). I enjoy reading the design of the analysis.

Concerns:

* The authors claim model degradation is the main cause of performance degradation of deep GNNs. However, what is model degradation is not defined formally.

* It seems the proposed Node Smoothing Level is a measure of **smoothness** instead of **over-smoothness**. Claiming over-smoothing is not a major issue using Node Smoothing Level is not very well-justified. I think a better metric should be used. I recommend the authors check Group Distance Ratio and Instance Information Gain proposed in [1].

* Why is the essential difference between the proposed “ResGCN” and “DenseGCN” in Figure 3 (b) and “ResGCN” and “DenseGCN” in [2].

* What is the difference between the findings of adding skip connections to GNNs in [2] [3] and the findings in Section 5.3? This should be discussed.

* It is unclear how the node-adaptive combination mechanism and residual connections between ET operations help with training deep GNNs. An ablation study needs to be carried out.

* What is the reason that the performance of DGMLP on ogbn-product is far from the leading methods on the OGB leaderboard?

* Why is the performance of baseline SIGN [4] is lower than the originally reported results (0.6568 ± 0.0006) on the leaderboard?

* What are the benefits of disentangled graph convolution over entangled graph convolution rather than scalability?

[1] Zhou, Kaixiong, et al. "Towards Deeper Graph Neural Networks with Differentiable Group Normalization." NeurIPS 2020.

[2] Li, Guohao, et al. "Deepgcns: Can gcns go as deep as cnns?." ICCV 2019.

[3] Li, Guohao, et al. "Deepergcn: All you need to train deeper gcns." arXiv 2020.

[4] Frasca, Fabrizio, et al. "SIGN: Scalable Inception Graph Neural Networks." arXiv 2020.

**Summary Of The Paper:**

This work performs a systematic study to analyze the main issues of the difficulty in training deep GNNs by disentangling the effects of embedding propagation (EP) and embedding transformation (ET). They find that the large $D_t$ is the root cause for the failure of deep GNNs. Node-adaptive combination mechanism and residual connections between ET operations are proposed to train deep GNNs.

**Summary Of The Review:**

(+) The systematic study of training deep GNNs of this work is interesting.

(-) Some claims are not well-justified. Some related works are not discussed properly. Alations need to be done for the proposed methods. The performance of DGMLP is not very strong.

---

> ### Author Response · Authors · 2021-11-24
> **Response to Reviewer-1, Part 1**
>
> Thanks for your constructive feedback! We believe that addressing this feedback will make our paper stronger.
>
> ### 1. Definition of Model degradation
> As we introduced in Section 4.2 in the original manuscript, model degradation is a phenomenon that the training and test accuracy firstly increases and then decreases rapidly when increasing the number of layers in one model, which is also defined in the ResNet paper[5].
>
>
> ### 2. The Proposed Node Smoothing Level
> Thanks for pointing out the related node smoothing metrics in [1]. We have read this paper in detail, and we observe that: 1) Group Distance Ratio makes classifies nodes of the same class label into a group (cluster), and it measures the ratio of inter-group distance over intra-group distance in the Euclidean space. 2) Instance Information Gain measures how much input feature information is preserved in the final representation.
>
> Group Distance Ratio cannot be applied in the unsupervised setting since the class information is unknown. Similar to Instance Information Gain, we consider the similarity between the input feature information and final representation in Eq.(5). Besides, we further propose to consider the similarity between the final representation and the representation at the stationary (over-smoothed) state. A smaller distance to the stationary state means the node feature is more likely to be over-smoothed. In other words, our proposed node smoothing level covers more information than the Instance Information Gain you mentioned.
>
>
> ### 3. Differences to DeepGCN and DeeperGCN
> The proposed "ResGCN" and "DenseGCN" in Figure 3 (b) are exactly the "ResGCN" and "DenseGCN" in [2], which are cited in the baseline part of Section 7.1. Sorry for the misunderstanding. We have added the citation of [2] in Section 2, where the "ResGCN" and "DenseGCN" are first introduced.
>
> **Findings of difference:**
> It should be emphasized that ResGCN and DenseGCN proposed in [2] is a coupled GNN model, and the residual and dense connections are applied among the GCN layers. Instead, these two skip connection mechanisms are directly used in the MLP part in our proposed DGMLP model.
>
> As we discussed in Section 4.2, although skip connections can lead to a small performance decline in extremely deep GCNs, the true reason why these two skip-connection mechanisms can work has not been well understood in [2]. Like most previous work, [2][3] simply blames the shallow architecture issue on problems including over-smoothing, over-fitting, and gradient vanishing without deep analysis. However, as we introduced in Section 4.1, Appendix C.1, and Appendix C.3, over-smoothing, over-fitting, and gradient vanishing are not the root causes of the performance degradation, which contradicts the findings proposed in [2].
>
> ### 4. The individual influence of node-adaptive combination mechanism and residual connections between ET operations.
> We have analyzed the influence of node-adaptive combination mechanism and residual connections, respectively, in Figure 6.
>
> **Node-adaptive combination mechanism.** As shown in Figure 6(a), we fix $D_t$ and increase $D_p$, and report the corresponding test accuracy on the ogbn-arxiv dataset when the propagation depth $D_p$ changes from 2 to 20. The experimental results show DGMLP consistently outperforms all the baseline methods when $D_p$ is greater than 6, which verifies the effectiveness of our node-adaptive combination mechanism.
>
> **Residual connections.** In Figure 6(b), we show the predictive accuracy of all the competing methods when $D_p$ is set to 10, and $D_t$ is increased from 1 to 10. The results show that the predictive accuracy of all the baseline methods, including SGC, DAGNN, and S$^2$GC decreases rapidly when $D_t$ becomes large, while the predictive accuracy of our DGMLP doesn't, which verifies the effectiveness of the residual connections in our approach.

---

> > ### Author Response · Authors · 2021-11-24
> > **Response to Reviewer-1, Part 2**
> >
> > ### 5. The performance of DGMLP and SIGN.
> > On the OGB leaderboard, SIGN[4] obtains the test accuracy of 65.68% by applying and integrating multiple filter matrices to smooth the node features, including standard normalized adjacency matrix, PPR diffusion matrix, and normalized triangle-induced adjacency matrix. For a fair comparison in Table 3, we only report the performance of SIGN with only the normalized adjacency matrix like DGMLP and all the other baselines. And it obtains the test accuracy of 64.2%, which is consistent with the results in SIGN's original paper[4] (the reported accuracy using the normalized adjacency matrix only is 64.28%).
> >
> > Most leading methods on the OGB leaderboard are not a single model. Instead, they are combined with several training tricks (e.g., C\&S and SLE), whether explicitly or implicitly. Besides, to better understand where the main performance gain comes from and the correctness of our findings, we seek a simple but effective design principle in DGMLP.
> > For example, the attention mechanism for node adaptive propagation is not adopted, and we just use a more interpretable and heuristic weighting mechanism in DGMLP. We can replace this mechanism with the attention mechanism proposed in SAGN [6] and get better performance. To sum up, the main contribution of the proposed DGMLP is not to outperform the current state-of-the-art baselines but to verify the correctness of our findings with respect to the over-smoothing and model degradation problems.
> >
> > ### 6. Benefits of disentangled graph convolution.
> > As we introduced in Section 5.1, under the disentangled graph convolution paradigm, the choices of the propagation depth $D_p$ and the non-linear transformation depth $D_t$ are more flexible as it breaks the limit that $D_p = D_t$ in the entangled graph convolution paradigm. As introduced in Appendix.B, this characteristic is important since the optimal $D_p$ and $D_t$ are different for different kinds of graph-structured data. Compared with the entangled graph convolution, the disentangled framework can adopt different values of $D_p$ and $D_t$ for optimal predictive accuracy.
> >
> > [1] Zhou, Kaixiong, et al. "Towards Deeper Graph Neural Networks with Differentiable Group Normalization." NeurIPS 2020.
> >
> > [2] Li, Guohao, et al. "Deepgcns: Can gcns go as deep as cnns?." ICCV 2019.
> >
> > [3] Li, Guohao, et al. "Deepergcn: All you need to train deeper gcns." arXiv 2020.
> >
> > [4] Frasca, Fabrizio, et al. "SIGN: Scalable Inception Graph Neural Networks." arXiv 2020.
> >
> > [5] He, Kaiming He, et al. Deep residual learning for image recognition. CVPR 2016.
> >
> > [6]  Sun, Chuxiong, at al. "Scalable and Adaptive Graph Neural Networks with Self-Label-Enhanced training." arXiv 2021.
> >
> > We are very glad to respond if you have any new questions.
> >
> > Respectfully,
> >
> > Paper753 Authors

---

> > > ### Comment · Reviewer_4Jnf · 2021-12-05
> > > **Response to authors**
> > >
> > > Thanks for your effort to address my questions. However, I am not fully convinced:
> > >
> > > 1. Definition of Model degradation. Yes, model degradation is mentioned in the ResNet paper. However, it is not defined formally. Could you formally define what is model degradation in GNNs? Why does the performance degenerate as the number of layers increases?
> > >
> > > 2.  Smoothness v.s. over-smoothness. Is the proposed Node Smoothing Level measuring over-smoothness or smoothness? If we are working in a supervised learning setting. Is Group Distance Ratio better than the proposed Node Smoothing Level?
> > >
> > > 3. Many methods on ogbn-product are using the entangle fashion of GNNs while achieving much higher performance, which makes me doubt the benefit of disentangled graph convolutions.

---

> > > > ### Author Response · Authors · 2021-12-06
> > > > **Response to Reviewer-1**
> > > >
> > > > Thanks for your continuous comments! We address your concerns as follows.
> > > > ### 1.Definition of Model degradation.
> > > > The phenomenon of model degradation is the same for both CNNs (e.g., ResNet) and GNNs, and we will define it formally in the revised manuscript.
> > > > ### 2. How does model degradation influence GNNs?
> > > > As introduced in Figure 1 of Section 2.2, GCN will degrade to MLP if $\hat{A}$ is the identity matrix, which is equal to removing the EP operation in all GCN layers. Due to the model degradation issue, we found both the training and test accuracy of MLP degenerates rapidly as the number of layers increases. As a result, if deep MLP cannot work well on graph data, how can we expect deep GNNs can work?
> > > > ### 3. Definition of smoothness
> > > > In our definition, smoothness is a metric that measures how the nodes are similar to their satisfactory state. Therefore, it can also be used to measure over-smoothness if the smoothness value (i.e., NSL in our paper) exceeds a pre-defined threshold. We think both the Group Distance Ratio and our proposed Node Smoothing Level are reasonable and meaningful. We will compare Group Distance Ratio and our proposed Node Smoothing Level in a supervised setting in the revised manuscript.
> > > >
> > > > ### 4.The benefit of disentangled graph convolutions.
> > > > We agree that “Many methods on ogbn-product are using the entangle fashion of GNNs while achieving much higher performance.” However, all the top-5 methods are disentangled. Besides, all the top-4 methods in ogbn-papers100M and ogbn-mag are disentangled, showing the large benefit of disentangled graph convolutions.
> > > >
> > > > We are very happy to respond if you have additional comments or concerns about our responses.
> > > >
> > > > Respectfully,
> > > >
> > > > Paper753 Authors

---

### Decision · Program_Chairs · 2022-01-20

**Decision:**

Reject

**Comment:**

The paper studies why existing deep GCNs suffer from poor performance and propose DGMLP to improve over existing GCNs. However, the reviewers think there are still many unjustified claims and the paper. Further, several reviewers question about the novelty of the proposed method, which seems to be a combination of existing approaches.

I suggest the authors to revise the paper by defining terms like model degradation and smoothness mathematically and try to justify each claim (e.g., the effect of disentangling) with solid experiments. These will significantly improve the analysis part and make the conclusions stronger.